# Intravenous Acetaminophen Does Not Provide Adequate Postoperative Analgesia in Dogs Following Ovariohysterectomy

**DOI:** 10.3390/ani11123609

**Published:** 2021-12-20

**Authors:** Jessica Leung, Thierry Beths, Jennifer E. Carter, Richard Munn, Ted Whittem, Sebastien H. Bauquier

**Affiliations:** 1Translational Research and Clinical Trials (TRACTs), U-Vet, Melbourne Veterinary School, The University of Melbourne, Melbourne 3030, Australia; jessica.leung2@unimelb.edu.au (J.L.); jennifer.carter@unimelb.edu.au (J.E.C.); bauquier@unimelb.edu.au (S.H.B.); 2Cognosco, Anexa Vet Services, Morrinsville 3110, New Zealand; rmunn@anexa.co.nz; 3College of Public Health, Medical and Veterinary Sciences, James Cook University, Townsville 4811, Australia; ted.whittem@jcu.edu.au

**Keywords:** canine, acetaminophen, ovariohysterectomy, postoperative, pain, plasma concentration

## Abstract

**Simple Summary:**

Acetaminophen is the most commonly used pain relief (analgesic) agent in humans worldwide and its use is becoming more frequent in dogs. However, limited evidence supports this use. This study aimed to investigate the analgesic effect of acetaminophen when administered as an intravenous injection post-operatively in female dogs. A total of 34 dogs were randomly divided into two groups and either administered acetaminophen or saline intravenously immediately after desexing. The dogs had their pain levels evaluated at 10, 20, 40, 60, 120, and 180 min after awakening from general anesthesia and the pain levels between groups were compared. Concurrently, the dogs had blood collected at 2, 5, 10, 40, and 80 min following injection of the acetaminophen. The blood was analyzed to quantify the levels of acetaminophen in the body. This study found that acetaminophen was no better than saline in providing analgesia in dogs following surgery. This study suggests that acetaminophen used alone may not be an appropriate post-operative analgesic agent for desexing procedures.

**Abstract:**

(1) Objective: To investigate the analgesic effects of intravenous acetaminophen after intravenous administration in dogs presenting for ovariohysterectomy. (2) Methods: 14 ASA I client-owned female entire dogs. In this randomized, blinded, clinical study, dogs were given meperidine and acepromazine intramuscularly before induction of anesthesia with intravenous propofol. Anesthesia was maintained with isoflurane in oxygen. Intravenous acetaminophen 20 mg/kg or 0.9% NaCl was administered postoperatively. Pain assessments were conducted using the Glasgow Pain Scale short form before premedication and at 10, 20, 60, 120, and 180 min post-extubation or until rescue analgesia was given. The pain scores, times, and incidences of rescue analgesia between the groups was compared. Blood was collected before and 2, 5, 10, 20, 40, and 80 min after acetaminophen administration. Acetaminophen plasma concentration was quantified by liquid chromatography-mass spectrometry. The acetaminophen plasma concentration at the time of each pain score evaluation was subsequently calculated. (3) Results: There was no significant difference in pain scores at 10 min, highest pain scores, or time of rescue analgesia between groups. In each group, 3 dogs (43%) received rescue analgesia within 20 min. (4) Conclusions: Following ovariohysterectomy in dogs, there was no detectable analgesic effect of a 20 mg/kg dosage of intravenous acetaminophen administered at the end of surgery.

## 1. Introduction

Acetaminophen is a non-opioid analgesic that is widely used in humans for its analgesic and antipyretic effects [1]. In the human population, the use of acetaminophen as an analgesic has been described since the 1950s. Currently, it is the first step in acute pain management along with aspirin in the World Health Organization analgesic ladder [1,2,3].

Acetaminophen’s precise mechanism of action is unknown, although it is usually hypothesized that acetaminophen works by inhibiting central COX receptors to elicit analgesia [3,4,5,6]. In children, an intravenous dose of acetaminophen rapidly enters the cerebral spinal fluid, and it is suspected that diffusion into the central nervous system plays a significant role in providing analgesia [7]. Despite its proven action on COX receptors, any possible anti-inflammatory actions of acetaminophen are weak [4]. Recently, other mechanisms of action of acetaminophen have been proposed such as an action on cannabinoid receptors [8] and serotonergic pathways to induce analgesia and prevent hyperalgesia in humans and rodents [5,9].

Acetaminophen is most commonly administered as an oral analgesic resulting in a delayed onset of action and lower plasma levels compared to intravenous administration [10]. When administered orally, a hysteresis loop has been documented, with peak analgesia occurring about an hour following peak plasma concentrations in humans [11]. In humans however, the popularity of intravenous acetaminophen is growing for acute post-operative pain. When given intravenously, the onset of analgesia occurs within 5 min [12]. Comparatively, the use of intravenous acetaminophen in animals is limited. Its use has been reported in dogs [13] and in horses [14] with variable analgesic efficacy.

In dogs, the pharmacokinetics of intravenous acetaminophen have been described in beagles, Spanish sighthounds and Labradors at doses of 10 mg/kg and 20 mg/kg [15]. Neither of the doses produced any deleterious effects on clinical biochemistry values. To the authors’ knowledge, the pharmacodynamics of intravenous acetaminophen in dogs in relation to the achieved plasma concentrations are yet to be investigated. Furthermore, it is unknown whether plasma concentrations of paracetamol pose any relevance to the tissue effect site.

This study aimed to investigate the analgesic effects of intravenous acetaminophen as an analgesic for acute post-operative pain in dogs. A secondary aim of this study was to correlate the analgesic effects of intravenous acetaminophen as a function of the plasma acetaminophen concentrations. Based upon previously published pharmacokinetic data, we hypothesized that intravenous acetaminophen would provide analgesia for at least an hour postoperatively in dogs [15] undergoing ovariohysterectomy.

## 2. Materials and Methods

### 2.1. Study Design

The study was a prospective, randomized, blinded, clinical trial. The study was approved by the University of Melbourne Animal Ethics Committee (number 1914886.2) and written informed owner consent was obtained before enrollment.

### 2.2. Animals

Client-owned female entire dogs weighing between 5 and 40 kg and presenting to the U-Vet Animal Hospital for ovariohysterectomy were eligible for inclusion. Exclusion criteria included dogs under 24 weeks of age, having more than one surgical procedure performed, an American Society of Anesthesiologists (ASA) status greater than I, and a body condition score of less than 3/9 or greater than 8/9. All animals were deemed healthy based on physical examination and basic biochemistry.

Dogs were randomly assigned to either the control group (S) or acetaminophen group (A) using the random function of a commercially available software package (Microsoft Excel 2016, Microsoft Corporation, Redmond, WA, USA).

### 2.3. Anesthesia and Surgery

Animals were premedicated with 7 mg/kg of meperidine (Pethidine Hydrochloride, DBL, Vic, Australia) and 0.02 mg/kg of acepromazine (A.C.P.2, Ceva Animal Health, Glenorie, NSW, Australia) administered intramuscularly into the cervical or epaxial muscles. Once sufficiently sedated, an intravenous catheter was placed into a cephalic vein for the administration of drugs and intravenous fluids (Compound Sodium Lactate, Fresenius Kabi Australia, North Ryde, NSW, Australia). Animals were anesthetized with intravenous propofol (Fresofol 1%, Fresenius Kabi Australia, North Ryde, NSW, Australia) titrated until endotracheal intubation could be achieved. Once intubated, animals were connected to an anesthetic machine through a rebreathing circuit. General anesthesia was maintained with isoflurane (Isoflo, Zoetis, Rhodes, NSW, Australia) in 100% oxygen. The isoflurane vaporizer was adjusted to maintain an anesthetic depth that was sufficient for surgery. Following induction of general anesthesia, a second intravenous catheter was placed in the contralateral cephalic vein for blood sampling as described below. If animals showed movement during surgery, 1 mg/kg propofol was administered intravenously.

All animals were monitored using a multiparametric monitor (Carescape Monitor B450, GE Healthcare, Paramatta, NSW, Australia) which included pulse oximetry, capnography, electrocardiography, oscillometric blood pressure, and esophageal temperature. Heart rate and respiratory rate were monitored from the pulse oximeter and capnograph, respectively. The anesthetist was unaware of the group allocation. A routine midline ovariohysterectomy was performed by a qualified veterinary surgeon, on some occasions assisted by a final year veterinary student. The surgical time was limited to a maximum of 1 h.

The time from premedication to induction, total surgical time, and total anesthetic time were recorded.

### 2.4. Acetaminophen Administration and Blood Sampling

The acetaminophen and placebo were prepared in a separate room by an investigator not involved in the pain scoring or plasma analysis, to maintain blinding. Animals in group A received 20 mg/kg acetaminophen (Paracetamol IV, Pfizer Australia, Sydney, NSW, Australia) whilst animals in group S received an equivalent volume of 0.9% saline (Sodium chloride 0.9%, Fresenius Kabi Australia, North Ryde, NSW, Australia). The acetaminophen product was a human intravenous preparation not licensed for veterinary use. The drugs were drawn up in one or two syringes depending on volume and the syringes wrapped in a cohesive elastic bandage to hide the differing colors of the two solutions. The wrapped syringes were handed to the primary investigator after the animal was moved into surgery.

At the completion of the surgery, animals were placed into lateral recumbency and administered the placebo or acetaminophen as an intravenous bolus over 30 s. Blood was sampled through the second intravenous catheter just prior to drug administration (time 0) and at 2, 5, 10, 20, 40, and 80 min following drug administration. For each sample, 0.5 mL of blood was collected and discarded as a scavenge. Immediately following this, 1 mL of blood was sampled and stored in a lithium heparin blood tube and the catheter flushed with heparinized saline. Blood sampling was performed irrespective of the group allocation to maintain blinding.

Blood samples were stored at 4 °C until all seven samples for each animal were collected. Within 1 h of collection of the final sample, all samples were centrifuged at 21 °C at 600 G-force for 15 min. The plasma was aspirated, transferred into microcentrifuge tubes, and stored at −80 °C until analysis for acetaminophen concentration. The samples were stored for a maximum of 6 months prior to analysis.

### 2.5. Pain Assessment

Isoflurane was discontinued 2 min after acetaminophen or saline administration. Dogs were extubated once swallowing and immediately transferred to a recovery ward for the remainder of the study. Animals were monitored for signs of postanesthetic dysphoria by two trained anesthesia personnel following extubation and during transport to the wards. Any animal showing signs of dysphoria including but not limited to vocalization, paddling, or nystagmus were excluded from analysis. Each dog was pain scored using the Glasgow Composite Measure Pain Score Short Form (intra-observer reliability intra-class correlation coefficient (ICC) 0.85; inter-observer reliability ICC 0.95 [16]) before anesthesia (pre-operative pain score) and at 10, 20, 60, 120, and 180 min after extubation. Those that received a pain score greater than five if non-ambulatory or greater than six if ambulatory were deemed in need of rescue analgesia. The intervention points were in line with those defined by Reid [17] who reported a linear discrimination and cross-validation of 0.86 for the rescue thresholds. Sedation was not assessed post-operatively.

Rescue analgesia consisted of methadone (Methone, Ceva Animal Health Pty Ltd., Glenorie, NSW, Australia) 0.2 mg/kg IV and meloxicam (Metacam, Boehringer Ingelheim Animal Health Australia, Macquarie Park, NSW, Australia) 0.2 mg/kg SQ. Animals were pain scored again 20 min following the administration of rescue analgesia to ensure adequate analgesia was achieved. If animals exceeded the rescue threshold at this time point, they were administered an additional 0.2 mg/kg methadone IV. Animals were pain scored again 20 min following the repetition of rescue analgesia. Data obtained after the provision of rescue analgesia were excluded from analysis.

The pain scores at each time point, the need for rescue analgesia, the time of rescue analgesia, and the amount of rescue analgesia required for each animal were recorded.

### 2.6. Plasma Acetaminophen Analysis

The determination of the acetaminophen plasma concentration was performed using liquid chromatography/mass spectrometry, by modifying the method described previously by Kam et al. [18]. Briefly, the process was as follows: Analytical standard acetaminophen (Acetaminophen European Reference Standard, Sigma Aldrich Pty. Ltd., Castle Hill, NSW, Australia) was used as a reference standard, and acetaminophen D4 (Acetaminophen D4, Cayman Chemicals, Ann Arbour, MI, USA) was used as the internal standard. A stock solution of 200 ng/mL acetaminophen-D4 was prepared in HPLC grade acetonitrile. Plasma was thawed at 4 °C.

A validation was performed using canine plasma as a matrix. Plasma calibrators were prepared by the addition of analytical standard acetaminophen to blank canine plasma. Using a stock solution of acetaminophen, a calibration curve was performed within the range of 0.5 μg to 30 μg/mL.

Prior to analysis, 20 μL of plasma sample was added to 320 μL of 200 ng/mL acetaminophen-D4 in a 1.5 mL microcentrifuge tube. The samples were vortexed for 2 min to mix and centrifuged at 600 G for 10 min. The supernatant was added to 180 μL of MilliQ water in a 96-well plate and the plate was vortexed for 10 min. Each 96-well plate contained 10 quality controls with the following concentrations made using serial dilutions of an acetaminophen stock solution: 0 μg/mL, 0.5 μg/mL, 1 μg/mL, 2 μg/mL, 5 μg/mL, 10 μg/mL, 15 μg/mL, 20 μg/mL and 30 μg/mL.

A sample volume of 10 μL was injected into the LCMS system. The LCMS system consisted of a Shimadzu LCMS 8050 including an autosampler, solvent pumps, column oven chamber, and triple quadrupole mass spectrometer (Shimadzu Australia, Rydalmere, NSW, Australia).

Acetaminophen was separated from plasma matrix on a C18 Poroshell 120 SB 2.1 mm × 50 mm 2.7 micron column (Agilent Technologies, Mulgrave, VI, Australia) at a flow rate of 0.2 mL/min. A gradient mobile phase was used, starting at 95% mobile phase A of 0.1% formic acid in MilliQ water and 5% mobile phase B of 100% HPLC grade methanol. The total run time was 6 min. Detection and quantification of acetaminophen were conducted using multiple reaction monitoring of acetaminophen (*m*/*z* 111.1) and the internal standard acetaminophen d4 (*m*/*z* 114.1) at the retention time of 1.0 min.

Where the acetaminophen quantification exceeded the upper limits of the calibration curve (i.e., was greater than 30 μg/mL), a subsequent dilution was performed on those samples. A stock diluent solution was made by adding 320 μL acetaminophen D4 (200 ng/mL) to 20 μL of acetonitrile and 3.06 mL of MilliQ water. A 20 μL aliquot of the previously prepared plasma sample was added to 180 μL of the diluent solution. As for the previous samples, 10 μL of this final solution was injected into the LCMS.

### 2.7. Statistical Analysis

An a priori power study was performed on the need for rescue analgesia between the groups. It was assumed that similar therapeutic plasma concentrations achieved in humans would provide analgesia in dogs. As such, based on previously published pharmacokinetic data [15], it was estimated that 5% of animals would need rescue analgesia in group P and 50% of animals in group S would require rescue analgesia within the first hour. A sample size of 28 (14 animals per group) would be required to demonstrate this difference with a power of 0.8 and alpha of 0.05.

Data were assessed for normality using a Shapiro—Wilk test. Comparison of weight, age, the time between premedication and induction, total anesthesia time, and total surgery time was analyzed with a 2-tailed Student’s *t*-test. Pre-operative pain scores, highest pain scores, time of rescue analgesia, and the time point evaluation scores were analyzed via a 2-tailed Mann—Whitney exact test. The need for rescue analgesia was analyzed using a Fisher’s exact test. Results are reported as mean ± SD or median (range). A *p*-value < 0.05 was deemed significant.

## 3. Results

Of the 15 dogs (1 Staffordshire Bull Terrier, 3 Pharoah Hounds, 1 mongrel dog, 1 Belgian Shepherd, 1 Lagotto, 2 Japanese Spitz, 1 Brittany Spaniel,1 Rottweiler, 2 Border Collies, 1 Golden Retriever) that completed the study, one animal (the Lagotto) was excluded due to a body condition score outside the prescribed inclusion criteria. The remaining 14 dogs were equally distributed between group S (*n* = 7) and group A (*n* = 7). No animal exhibited signs of dysphoria on recovery from anesthesia. The study was terminated prematurely due to a high prevalence of rescue analgesia. In both groups, three out of seven dogs required rescue analgesia after the 20 min pain score, and in both groups, a total of four out of seven dogs required rescue analgesia after the 60 min pain score. At the end point of the study (180 min), four out of seven animals in group A required rescue analgesia, whilst six animals in group S required rescue analgesia. Furthermore, meperidine became unavailable in the country where the study took place, precluding additional enrolment using the same anesthesia protocol.

There was no significant difference in age (*p* = 0.59) or weight (*p* = 1.0) between the groups. There was no significant difference between the time between premedication and induction (*p* = 0.48), total surgical time (*p* = 0.45), or total anesthesia time (*p* = 0.70) between the two groups. The pharmacodynamic data are summarized in Table 1 and Table 2.

There was no significant difference in the pre-operative pain scores between the two groups (*p* = 0.70) or at any time point following extubation. There was no significant difference between the highest post-operative pain score between group S and group A (*p* = 0.78). There was no significant difference between the need for rescue analgesia (*p* = 1.0) or time of rescue analgesia (*p* = 0.46) between group S and group A. Two animals required a second dose of rescue analgesia: one in group A and one in group S.

The lower limit of detection for the LCMS was 0.048 μg/mL. The inter-assay variability coefficient was 1.4% and intra-assay variability coefficient was 1.7%., the mean accuracy was 105.2%, and mean was precision 98.3%.

The plasma concentrations of acetaminophen are shown in Table 3.

## 4. Discussion

When administered intravenously, acetaminophen showed no detectable additional analgesic effects following ovariohysterectomy in dogs premedicated with meperidine and acepromazine. Furthermore, neither meperidine alone nor meperidine and acetaminophen provided reliable post-operative analgesia.

Acetaminophen is used commonly as a post-operative analgesic in children and adults, either as a sole analgesic or in combination with opioids for mild to moderate pain. In cases of moderate to severe pain post-operatively, a Cochrane review found a positive result with acetaminophen administration in only up to 36% of patients [19]. Multiple studies in humans do not endorse the use of intravenous acetaminophen as a sole analgesic, particularly in abdominal surgery [20,21]. When used in conjunction with opioids, acetaminophen decreases the requirements of patient-controlled analgesia although it may not cause significant differences in pain scores between groups [22,23,24]. In patients experiencing mild to moderate pain, acetaminophen produces more consistent analgesia than in patients with severe pain [25].

The duration of the analgesic effects of meperidine has been reported to be dose-dependent [26,27]. A 2 mg/kg IM dose provided analgesia for up to 90 min, and 3 mg/kg IM dosage produced analgesia for 120 min in dogs undergoing castration [26]. A further study using a 5 mg/kg IM dose in dogs showed acceptable analgesia for up to 4 h following ovariohysterectomy [27]. Conversely, Lascelles et al. [28] administered 5 mg/kg IM to dogs either within 20–30 min prior to ovariohysterectomy or immediately after. Dogs administered post-operative meperidine had significantly lower pain scores than animals administered pre-operative meperidine for one hour post-extubation. However, in the same study, animals administered pre-operative meperidine had lower mean pain scores across the time span of the entire study as well as lower mechanical nociceptive thresholds, indicating that meperidine provides short-lived analgesic action but prevents central sensitization of pain if given pre-emptively [28]. Lascelles et al. [28] speculate whether a higher dose would provide longer lasting analgesia, although recommend against it. In the present study, all animals received meperidine as part of their premedication. However, in the control group of the present study, a pre-operative dose of 7 mg/kg IM did not reliably provide adequate analgesia following ovariohysterectomy. Even with the putative additional analgesic effect from acetaminophen administered at the end of the surgery, the analgesic requirements of the ovariohysterectomy were not met. In dogs, an ovariohysterectomy is classified as producing moderate pain [29]. Vettorato and Bacco [27] used the Colorado Pain Scoring System in their study for which an agreed rescue threshold has not been determined, unlike the Glasgow Composite Measure Pain Scale- Short Form used in this study. Vettorato and Bacco [27] also set an arbitrary rescue threshold of 13, which may have affected their results and thus could account for the different results between their study and the present study. Furthermore, castration is deemed to cause a lower degree of pain than ovariohysterectomy, which may also account for the difference between this study and the results by Waterman & Kalthum [26]. The lack of anti-nociception when acetaminophen is administered alone is also reported in other species [14,21]. In horses, an acetaminophen intravenous infusion alone did not cause any increase in the anti-nociceptive thresholds, although an increase in anti-nociception was achieved when combined with intravenous tramadol infusions [14].

The timing of acetaminophen administration may affect its efficacy. Several human trials show a decrease in post-operative pain scores and reduced opioid requirement when intravenous acetaminophen is administered before the start or before the end of surgery compared to controls, including for those undergoing abdominal surgery [30,31]. It is hypothesized that acetaminophen exerts a central effect, potentially through serotonin or cannabinoid pathways, and prevents the central sensitization of pain to elicit a post-operative analgesic effect [32]. However, a comparison of acetaminophen administered intravenously either prior to surgery or once conscious in the recovery ward for children having strabismus surgery showed no difference in pain scores between the two groups [33].

When administered 30 min prior to ovariohysterectomy incision in dogs, a 15 mg/kg dose of acetaminophen IV administered every 8 h was as effective as 4 mg/kg carprofen IV or 0.2 mg/kg meloxicam IV administered every 24 h for post-operative analgesia for 48 h [13]. In that study, all dogs also received an intra-operative fentanyl constant rate infusion, and pain assessments were performed using a Dynamic Interactive Visual Analog Scale (DIVAS) and Pain Scale of the University of Melbourne (UMPS) scoring systems. Those results contradict the present pain score results which may again be attributed to the difference in pain scoring schemes. Furthermore, the animals in the Hernandez-Avalos et al. study were administered an intraoperative fentanyl infusion which may have influenced the post-operative period by exerting an anti-hyperalgesic effect, as is seen with other opioids [34]. It is unlikely the fentanyl itself would have persisted into recovery due to its short terminal half-life.

The absorption of intramuscular meperidine in conscious dogs is widely variable, with some animals displaying a biphasic absorption [26]. Individual variability in absorption may result in a prolonged duration of action and may, at least partially, be responsible for the large range of post-operative pain scores encountered. A concurrently performed meperidine assay would be required to discount the individual variability of meperidine absorption and its effect.

The therapeutic plasma concentration of acetaminophen required to produce analgesia in humans is deemed to be between 10 to 20 μg/mL [11]. In this study, this threshold was exceeded in all animals after administration of 20 mg/kg acetaminophen IV. However, for all dogs in this study, acetaminophen was below this threshold by 40 min post-drug administration, with the majority below the range by 20 min. For most study participants, this time frame coincided with the 10 and 20 min post-extubation pain scores. Based upon the data obtained in this study, and correlated with other published pharmacokinetic studies, it is feasible that an intravenous dose of acetaminophen does not produce sufficiently sustained plasma concentrations to produce clinically relevant analgesia. Alternatively, the necessary plasma concentrations of acetaminophen for analgesia in dogs may be higher than reported in humans.

A significant limitation of the present study is the small sample size. However, due to the large number of animals that required rescue analgesia in both groups, it was deemed unethical to continue the study to reach the large numbers required for significance. It is possible that the small sample size, further compounded by excluding animals from the study as they required rescue analgesia resulted in a type II error.

A further limitation of this study is the upper limit of calibration for the LCMS was 30 μg/mL. This necessitated a subsequent dilution of the samples as many samples were higher than this concentration. This may have increased the variation and potential sources of error in the analysis.

## 5. Conclusions

In the present study, the authors were unable to demonstrate any additional analgesic effects of intravenous acetaminophen (20 mg/kg IV), when compared to saline, when administered following ovariohysterectomy in dogs premedicated with meperidine and acepromazine. Furthermore, within the current study conditions, neither meperidine (7 mg/kg IM preoperatively) alone or in combination with acetaminophen provided reliable analgesia post ovariohysterectomy.

## Figures and Tables

**Table 1 animals-11-03609-t001:** Age, weight, total anesthesia time, total surgical time, time between drug administration and first pain score, highest pain scores, time of rescue analgesia, and number of rescue analgesia doses for dogs receiving a placebo or 20 mg/kg acetaminophen IV following ovariohysterectomy. There was no significant difference between age, weight, total anesthesia time, total surgical time, time between drug administration and first pain score, highest pain scores, time of rescue analgesia and number of dogs requiring rescue analgesia doses between the groups. Data are reported as mean± SD or median (range).

Data	Acetaminophen Group	Saline Group	*p*-Value
Age (years)	1.1 ± 0.62	2.0 ± 1.4	0.59
Weight (kg)	18.7 ± 6.7	20.9 ± 1.8	1.0
Time between premedication and induction (min)	27 ± 6	23 ± 9	0.48
Total anesthesia time (min)	78 ± 23	76 ± 15	0.70
Total surgical time (min)	44 ± 17	37 ± 11	0.45
Time between drug administration and first pain score (min)	20 ± 2	20 ± 3	0.75
Highest pain scores	9 (3–13)	8 (6–9)	0.78
Time of rescue analgesia (min)	40 (20–180)	40 (20–120)	0.46
Number of dogs requiring rescue analgesia	4	6	1.0

**Table 2 animals-11-03609-t002:** Pre-operative and post-operative pain scores for dogs receiving saline or 20 mg/kg acetaminophen following ovariohysterectomy. Animals were pain scored at 10, 20, 60, 120, and 180 min after extubation. Animals that received rescue analgesia were excluded from subsequent data analysis. There was no significant difference between the dogs receiving saline or acetaminophen at any time post-extubation. The number of animals included at each time point is indicated in [*n*]. Data are presented as median (range).

Time of Pain Score	Acetaminophen Group	Saline Group
Preoperative	0 (0–4) [7]	0 (0) [7]
10 min	3 (1–13) [7]	2 (1–9) [7]
20 min	3 (2–9) [5]	3 (2–9) [5]
60 min	4 (1–9) [4]	5.5 (3–8) [4]
120 min	5 (2–6) [3]	7 (5–9) [3]
180 min	10 (1–11) [3]	5 (–) [1]

**Table 3 animals-11-03609-t003:** Plasma concentrations achieved for dogs administered 20mg/kg acetaminophen intravenously following ovariohysterectomy. Plasma concentrations were determined using LCMS.

Time (mins)	Acetaminophen Plasma Concentrations (ug/mL)							
	Dog 1	Dog 2	Dog 3	Dog 4	Dog 5	Dog 6	Dog 7	Dog 8
2	65.826	29.399	43.883	53.971	37.764	36.854	41.061	33.234
5	28.464	22.742	27.200	27.130	25.562	20.983	26.553	18.490
10	18.527	14.545	16.638	16.487	14.900	12.155	15.682	16.190
20	13.409	13.219	12.929	12.421	9.231	7.785	8.210	12.857
40	9.791	9.046	7.090	9.787	4.192	5.827	6.021	9.669
80	6.448	6.492	4.294	4.705	2.685	2.915	3.470	4.918

## Data Availability

The data presented in this study are available on request from the corresponding author. The data are not publicly available due to privacy.

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
