# Peer review of "Intravenous Acetaminophen Does Not Provide Adequate Postoperative Analgesia in Dogs Following Ovariohysterectomy"

_animals, 2021, doi:10.3390/ani11123609_

Round 1
Reviewer 1 Report
This article reports a study that is clearly underpowered but that was stopped due to the number of animals needing ‘rescue’ and the loss of the use of meperidine – it is not clear which of these was the real deciding factor. The authors measured plasma concentrations of acetaminophen, but this is not the effect site and it is not clear from the study or from the literature how long it takes for acetaminophen to reach a full effect after IV administration. In the introduction (line 59) the authors state that analgesia in children was seen within 5 minutes but the reference given at the end of this sentence has no mention of this time frame and the authors state that we don’t know the pharmacodynamics of acetaminophen in dogs. The drug was administered at the end of the surgery and the first pain score was done at 10 minutes after extubation so this is probably within 30 minutes of administration (values not given). If we don’t know the time of onset, the conclusion may not be that acetaminophen did not work but that it was not allowed enough time to have its full effect. The authors also state that meperidine became unavailable and ‘therefore’ the 10-minute pain score was the only one used for statistical analysis. I don’t understand the logic here – how did not having meperidine make it necessary to only test one time point? In the Results the authors state that ‘three out of seven and four out of seven dogs required rescue analgesia within 20 minutes and one hour respectively’ but in table 2 it appears as though only 2 animals in each group were rescued at 20 minutes and 3 at 60 minutes (according to the n). The authors also don’t mention that by 180 minutes 6/7 of the saline group had been rescued vs 4/7 of the acetaminophen group. Could that have been significant if enough animals had been studied?
The pain scale used, at least in my hands, has been very poor at differentiating pain from postoperative excitement in the early part of recovery. Scores can end up quite high e.g. a 3 for ‘nervous or anxious’, a 1 for ‘unsettled’, a 5 for ‘cry’ because it is already crying, would give a score of 9, but may not have anything to do with pain. The authors need to assure the reader that these scores were pain related and not related to dysphoria.
In the legends for the table the authors need to state what they are showing – I know it is stated in the statistics section, but the tables should be able to be understood on their own (both units and mean/median should be stated). In table 1 it looks as if the ‘time of rescue’ data are not normally distributed – if the SD exceeds the mean that usually suggests a problem – perhaps present these data as median and range? I am also guessing that the data are largely presented at a level that is more accurate than the measurement – were the times recorded to the second or just in whole minutes? A pain score of 7.8 is not possible. Please report the data to the accuracy of your measurement.
In table 3 you report acetaminophen concentrations at up to 6 decimal places. This implies that the limit of detection was in the pg/mL range? You should supply the limit of detection and limit of quantification as part of your statement about the analysis.
In the discussion the authors talk about the dose and analgesic effect of meperidine (line 262). Another reference that should be used here is: Post-operative central hypersensitivity and pain: the pre-emptive value of pethidine for ovariohysterectomy X B D Lascelles 1 , J P Cripps, A Jones, E A Waterman. Pain 1997 Dec;73(3):461-471. doi: 10.1016/S0304-3959(97)00141-3. This study shows that meperidine at 5 mg/kg only provided 1 hour of analgesia when administered immediately prior to recovery.
Line 281 – the authors state that acetaminophen did not increase nociceptive threshold – given the nature of the action of this drug you wouldn’t expect it to raise a threshold that doesn’t involve inflammation would you?
Line 304 – the fentanyl may not have persisted into the recovery but could it have had a pre-emptive effect as in the above reference from Lascelles?
Line 313 that should be µg/mL?
References – The style is inconsistent and several do not state the journal e.g 3, 8, 16, 27 and 30. Please check the journal style and conform to that.
Author Response
See attached document

Reviewer 2 Report
Overall comments:
There is a lack of information on the efficacy of acetaminophen (paracetamol) for managing acute pain in dogs and data is needed as it is a drug that is not scheduled / controlled and is available globally – certainly as an oral formulation. The intravenous formulation, however, is less commonly used in humans including children and rarely in dogs. As far as this reviewer knows there is no animal approved product (oral or intravenous) of acetaminophen alone. Pardale V is a combination of paracetamol plus codeine and is approved for use in dogs in some countries.
Unfortunately, this paper fall short on contributing a lot to our knowledge on this drug. This could perhaps be resubmitted as a short communication.
Specific comments:
The Abstract is misleading as it discusses when pain assessment were done (at 6 time points), then says there was no difference between the acetaminophen and saline group at 10 minutes: it is not until you read the whole paper you find out this was the only time point that a statistical comparison made. The final line of the abstract should state again that this conclusion that the drug was given after the end of surgery.
Line 62: this study was with ORAL acetaminophen PLUS codeine, not IV paracetamol.
Line 92: what was the basis for this dose of meperidine? (should be discussed somewhere)
Line 95: “sufficiently sedated” – was a sedation score used?
Line 119: would be helpful to say this is a human product and not authorized for use in dogs.
Section 2.5: it should be clarified that if dogs were rescued, they were considered treatment failures and not used for data analysis after rescue – you find this out later, but it can be included in this section.
Section 2.7: can the power and alpha be recalculated using the number of dogs that DID complete the study (n-14, not 28?).
Results:
The reasons for terminating the study are clear – drug shortage and high failure rate.
Table 2. it should be made clear on this table that times 20, 60, 120 and 180 were not statistically compared. Or do not include those times and describe in results when the dogs failed the treatment(s). More detail on which dog(s), which group and when (time) the highest scores were recorded would be a good addition.
Table 3 and Figure 1: use one or the other. The variability in plasma levels is more obvious in the Table as the Figure has a logarithmic scale.
Discussion:
Line 302: change to: “administered intraoperative fentanyl) or “administered an intraoperative fentanyl infusion”.
The timing (pre-operatively versus postoperatively) of administration needs more discussion. If NSAIDs are approved for pre-operative use, why was the decision made to give paracetamol after surgery – this is not really discussed.
References:
It would be worth considering this one: Lascelles BDX, Waterman AE, Cripps PJ, Livingston A, Henderson G. Central sensitization as a result of surgical pain: investigation of the pre-emptive value of pethidine for ovariohysterectomy in the rat. Pain. 1995 Aug;62(2):201-212. doi: 10.1016/0304-3959(94)00266-H. PMID: 8545146.
Reference 12; capitalize murrell
Author Response
See attached document

Reviewer 3 Report
The manuscript "Intravenous acetaminophen does not provide adequate postoperative analgesia in dogs following ovariohysterectomy" describes the analgesic effect of perioperative intravenous paracetamol in dogs undergoing ovariohysterectomy. The study is well designed with a concise manuscript and clear discussion of its limitations.
Minor comments:
The Glasgow composite measure pain scale is meant for dogs not being sedated. Assessements were close to end of anesthesia. Did you assess sedation in your dogs? please add this information.
For completenes, please indicate that you report median pain scores in Table 2 and log concentrations in the graph.
Author Response
See attached document

Round 2
Reviewer 1 Report
Line 47 – the COX-3 receptor mechanism has been called into question since it is not found in humans or rats. Perhaps use a more recent review of mechanisms e.g. Clin Exp Pharmacol Physiol. 2021;48:3–19. Paracetamol – An old drug with new mechanisms of action. Grzegorz W. PrzybyÅ‚a, Konrad A. Szychowski, Jan GmiÅ„ski.
Line 54 the role of acetaminophen in serotonergic pathways appears to depend on metabolites – do you know if those metabolites are produced in the dog?
Line 59 – use the original reference Br J Anaesth . 2005 May;94(5):642-8. doi: 10.1093/bja/aei109. Epub 2005 Mar 24. Onset of acetaminophen analgesia: comparison of oral and intravenous routes after third molar surgery P L Moller 1 , S Sindet-Pedersen, C T Petersen, G I Juhl, A Dillenschneider, L A Skoglund
Line 60 – wasn’t the hysteresis loop only with oral paracetamol?
Line 67 – I think it would be worth adding here that the plasma concentrations do not necessarily correlate with the activity at the site of action. Robenacoxib has a very short plasma half-life but stays at the effect site for 24 hours or more.
Line 126 – please state what IV bolus means – over 2, 20, 200 seconds?
Line 159/234/241 – data are plural
Line 180 – you only calibrated to 30 µg/mL – some of your plasma samples exceeded this concentration – this should be pointed out as a weakness of the study.
Line 222 – still no mention of the 4/7 vs 6/7
Line 223 – you still haven’t confirmed if this was the real reason the study was terminated
Table 1 -having found that the time of rescue analgesia data are non-parametric did you change your analysis?
Line 284 speculated
Line 322 - repetition
The references still don’t match the requirements for this journal e.g. Boissy, A.; Manteuffel, G.; Jensen, M.B.; Moe, R.O.; Spruijt, B.; Keeling, L.J.; Winckler, C.; Forkman, B.; Dimitrov, I.; Langbein, J.; et al. Assessment of positive emotions in animals to improve their welfare. Physiol. Behav. 2007, 92, 375–397.
Author Response
Dear Reviewer,
Please find attached the response to your advice and comments
Thank you
The Authors
